# In Vivo Transmigrated Human Neutrophils Are Highly Primed for Intracellular Radical Production Induced by Monosodium Urate Crystals

**DOI:** 10.3390/ijms21113750

**Published:** 2020-05-26

**Authors:** Lisa Davidsson, Agnes Dahlstrand Rudin, Felix Peter Sanchez Klose, Alicia Buck, Lena Björkman, Karin Christenson, Johan Bylund

**Affiliations:** 1Department of Rheumatology and Inflammation Research, Institute of Medicine, Sahlgrenska Academy at University of Gothenburg, 405 30 Gothenburg, Sweden; lena.i.bjorkman@vgregion.se; 2Department of Oral Microbiology and Immunology, Institute of Odontology, Sahlgrenska Academy at University of Gothenburg, 405 30 Gothenburg, Sweden; agnes.dahlstrand.rudin@odontologi.gu.se (A.D.R.); felix.klose@gu.se (F.P.S.K.); alicia.buck@gmx.de (A.B.); karin.christenson@gu.se (K.C.); johan.bylund@gu.se (J.B.)

**Keywords:** inflammation, gout, NETs, neutrophil extracellular traps, granule, NADPH-oxidase, myeloperoxidase, MPO

## Abstract

Gout is an inflammatory disease caused by monosodium urate (MSU) crystals. The role of neutrophils in gout is less clear, although several studies have shown neutrophil extracellular trap (NET) formation in acutely inflamed joints of gout patients. MSU crystals are known to induce the production of reactive oxygen species (ROS) and NET formation in neutrophils isolated from blood, but there is inconclusive knowledge on the localization of ROS production as well as whether the ROS are required for NET formation. In this report we demonstrate that MSU crystals activate human neutrophils to produce ROS exclusively in intracellular compartments. Additionally, in vivo transmigrated neutrophils derived from experimental skin chambers displayed markedly increased ROS production as compared to resting blood neutrophils. We also confirmed that MSU stimulation potently induced NET formation, but this response was not primed in in vivo transmigrated neutrophils. In line with this we found that MSU-triggered NET formation was independent of ROS production and proceeded normally in neutrophils from patients with dysfunctional respiratory burst (chronic granulomatous disease (CGD) and complete myeloperoxidase (MPO) deficiency). Our data indicate that in vivo transmigrated neutrophils are markedly primed for oxidative responses to MSU crystals and that MSU triggered NET formation is independent of ROS production.

## 1. Introduction

Gout is the most common inflammatory arthritis worldwide [1] and gouty attacks are evoked by oversaturation of uric acid that precipitates in and around the joints as needle-shaped monosodium urate (MSU) crystals. The inflammatory response in gout is initiated when MSU crystals are ingested by macrophages, leading to activation of the NALP3 inflammasome and subsequent production and release of active IL1β that triggers a massive inflammatory response with the recruitment of immune cells, predominantly neutrophils, from the bloodstream [2,3,4]. When neutrophils are recruited from blood to an inflamed tissue, they undergo a number of cellular changes and as a result they become hyper-responsive, or primed, and in many aspects distinct from the cells left in circulation [5]. As professional phagocytes, the principal task of neutrophils is to recognize and clear invading microbes and cell debris by phagocytosis, which involves uptake, killing and degradation of the prey. The ingested prey can be killed through a combination of proteolytic enzymes and toxins released into the phagosome, and the production of reactive oxygen species (ROS) [6]. ROS are formed by the help of an electron transporting enzyme, NADPH-oxidase, that can be assembled in the plasma membrane, as well as in membranes of intracellular vesicles, giving neutrophils the ability to direct ROS production to distinct subcellular sites, e.g., into the phagosome, or to the extracellular milieu [7]. In addition, intracellular ROS production may also take place in the total absence of phagosome formation and such ROS are likely formed in granules [7,8]. The importance of ROS production is seen in patients suffering from chronic granulomatous disease (CGD), a rare, inborn immunodeficiency associated with bacterial and fungal infections as well as a variety of inflammatory symptoms. Patients with CGD lack a functional NADPH-oxidase which results in the inability to produce ROS [9]. Neutrophils are also able to capture microbes extracellularly through the formation of neutrophil extracellular traps (NETs) [10,11,12,13]. NETs consist of DNA and granule proteins, e.g., proteases and anti-microbial peptides, that are being thrown out of the neutrophil to form sticky extracellular webs in which microbes can be captured. In vitro, NETs can be induced by many different stimuli, e.g., bacteria [14], fungal hyphae [15], immune complexes [16] and certain cytokines [14,17]. The best characterized example of NET formation describes a coordinated series of cellular events in which ROS production is a crucial step [18]. In vitro, the standard procedure to trigger NET formation by a ROS-dependent pathway, is to expose cells to the artificial stimulus phorbol ester PMA. We recently showed that the ROS crucial for PMA-triggered NET formation are those generated intracellularly in non-phagosomal compartments, likely resulting from the fusion of distinct granule types [19] and that these ROS needs to have been processed internally by myeloperoxidase (MPO) to drive the NET formation [20]. 

The interaction of neutrophils and MSU crystals has been rather extensively studied in vitro using cells isolated from peripheral blood. When such resting cells encounter MSU crystals they produce ROS [21,22,23,24,25,26] and they form NETs [4,22,25,26,27,28]. Some studies report that MSU-induced NET formation is dependent on ROS production [25,26], while others report that the two events are not mechanistically linked [22]. In gout, neutrophils are likely to encounter MSU crystals in tissues and not in circulation. Several studies have shown NETs in acutely inflamed joints [4,22,25,26,27], but knowledge on how tissue neutrophils react to MSU crystals is scarce. 

In this study we aimed to further investigate the interaction between MSU crystals and neutrophils, with special emphasis on in vivo transmigrated neutrophils isolated from tissues. We show that neutrophils produce intracellular, but not extracellular, ROS in response to MSU crystals and that the ROS response is potently increased in in vivo transmigrated as well as in in vitro primed neutrophils. We also demonstrate that neutrophils from blood as well as from tissues form NETs in response to MSU crystals. However, the NET formation in response to MSU crystals was not primed in tissue neutrophils. In line with this, we found that NET formation in response to MSU crystals was independent of ROS production and/or MPO activity.

## 2. Results

### 2.1. MSU Crystals Trigger Intracellular ROS Production in Neutrophils

ROS can be produced at distinct subcellular sites [7] and 300 µg/mL MSU crystals triggered a significant (*p* < 0.0001 compared to buffer treated cells, *n* = 16), robust and sustained production of intracellular ROS in neutrophils from peripheral blood as measured by luminol-enhanced CL (Figure 1A). The MSU response was similar to that induced by PMA, albeit not of the same magnitude (Figure 1A), and a clear dose-dependency was noted (Figure 1B). 

To instead measure extracellular release of ROS, we first used (isoluminol-amplified) extracellular CL. At higher doses (up to 500 µg/mL) of MSU crystals, no extracellular CL response was seen (Figure 1C) and the signal, in fact, seemed to dip below background levels (cells treated with buffer) (Figure 1C, inset). Samples stimulated with lower doses (<0.1 µg/mL) of MSU crystals were indistinguishable from buffer stimulated samples. The extracellular CL system detects superoxide anion specifically [29] and it is sensitive to antioxidants as well as to light-scattering particles that interfere with detection. We thus utilized a complimentary method to quantify extracellular ROS by the H_2_O_2_ specific probe Amplex red. Samples were stimulated for 20 min and then, to remove potentially light-scattering components, briefly centrifuged, before supernatants were analyzed. The MSU crystals (300 µg/mL) did not trigger extracellular H_2_O_2_ release and recorded levels were similar to buffer treated samples (Figure 1D). To summarize, MSU crystals trigger human neutrophils to produce intracellular, but not extracellular ROS.

### 2.2. The Oxidative Response to MSU Is Dependent on the NADPH-Oxidase

To ascertain that the MSU induced ROS stemmed from the NADPH-oxidase, we pretreated cells with two different pharmacological inhibitors of this enzyme before stimulation with MSU crystals: diphenyleneiodonium (DPI), a widely used, but rather unspecific inhibitor of flavoproteins, and GSK2795039 (GSK), a quite specific inhibitor of the phagocyte NADPH-oxidase [30,31]. Both inhibitors completely blocked MSU-induced intracellular ROS production (Figure 2A). Furthermore, neutrophils from one patient with chronic granulomatous disease (CGD; an inborn disease with a non-functional NADPH-oxidase; [9]) did not produce any ROS upon stimulation with either PMA (not shown) or MSU crystals (Figure 2B).

### 2.3. The Oxidative Response to MSU Can Be Inhibited by Colchicine and Cytochalasin B

Intracellular ROS production in neutrophils can take place in phagosomes, but also in granules independently of particle ingestion [7,8]. Phagocytosis is dependent on actin polymerization and when actin polymerization was blocked by cytochalasin B, intracellular ROS production in response to MSU or opsonized *Staphylococcus aureus* was inhibited while the PMA response was not affected (Figure 3A). This indicates that the ROS production induced by MSU crystals is dependent on a phagocytic process. Further, colchicine (a drug that inhibits microtubule polymerization and that is used to treat gout) also decreased MSU-induced ROS production (Figure 3B). Interestingly, colchicine seems to specifically affect MSU-induced ROS production and it did not dampen intracellular CL triggered by PMA or *S. aureus* (Figure 3B).

### 2.4. Increased MSU Responses after In Vivo Transmigration or In Vitro Priming

Neutrophil transmigration from blood to tissue is typically associated with a number of cellular changes. Such changes, e.g., exposure of receptors necessary for attachment and chemotaxis, occur mainly as a result of granule mobilization, whereby granule-localized receptors become exposed and ready to react to external stimuli [32]. Hence, tissue neutrophils are typically primed, or hyper-responsive, and in many aspects distinct from the cells left in circulation [5]. The oxidative response of in vivo transmigrated neutrophils from skin chambers to MSU crystals was potently enhanced as compared to the response from peripheral blood neutrophils from the same individuals (Figure 4A). In a similar manner, blood cells subjected to in vitro priming with TNFα or IL-1β were also hyper-responsive to MSU crystals and produced elevated levels of intracellular ROS upon stimulation (Figure 4B). Neither TNFα, nor IL-1β, activate the NADPH-oxidase per se (data not shown), but they clearly primed cells for subsequent activation by MSU crystals.

We have recently shown that in vivo transmigrated neutrophils derived from synovial fluid of patients with inflammatory (non-gouty) arthritis, in striking contrast to the highly primed phenotype of neutrophils from skin models, often display very limited signs of priming [33]. In line with these findings we find that neutrophils from (non-gouty) synovial fluid display varying levels of priming with respect to the oxidative response to MSU crystals (Appendix A).

### 2.5. MSU Crystal Induced NET Formation is ROS Independent

As reported previously [4,22,25,26,27], we found that stimulation of peripheral blood neutrophils with MSU crystals resulted in NET formation with widespread extracellular DNA strands decorated with MPO (Figure 5A). NET formation was quantified using Sytox green staining for extracellular DNA (Figure 5B); MSU crystals (300 µg/mL) triggered a statistically significant DNA release (*p* = 0.0005 compared to buffer treated cells after 180 min, *n* = 12) and the response was dose dependent with effects at similar concentrations as those triggering ROS production (Figure 5B). At high doses (>300 µg/mL) NET formation was more rapid, and reached higher levels, than when triggering NET formation with PMA (Figure 5B). 

NET formation in response to PMA is critically dependent on ROS production, and on the ability of MPO to process ROS intracellularly [20]. Several conflicting reports [22,25,26] make the role of ROS in MSU-triggered NET formation unclear. That stimulation with MSU crystals results in intracellular CL is evidence that MSU crystals trigger ROS production and MPO processing intracellularly. In our hands, NET formation following stimulation with MSU crystals was not affected by the presence of the NADPH-oxidase inhibitors DPI or GSK (Figure 6A), used at doses that completely inhibit intracellular ROS production (Figure 2A and [31]). In addition, neutrophils from one CGD patient as well as from one individual with complete MPO-deficiency [34] were perfectly able to form NETs in response to MSU crystals, but not to PMA (Figure 6B). These data strongly support the view that MSU triggered NET formation is not dependent on ROS production by the NADPH-oxidase and/or MPO.

### 2.6. In Vivo Transmigrated Neutrophils Are not Primed for NET Formation

In vivo transmigration typically prime neutrophils regarding ROS production, and this was also seen for MSU stimulation (Figure 4). Not much is known whether such priming also relate to NET formation. When stimulated with MSU crystals, skin chamber neutrophils (Figure 7A), as well as tissue neutrophils derived from synovial fluid (Figure 7B), formed NETs in a very similar manner, both regarding kinetics and amplitude, as did blood neutrophils from the same donor assayed in parallel. These data fit well with our findings that MSU-triggered NET formation is independent of ROS production. Similar findings are obtained when using PMA as a NET-trigger (Figure 7A,B). These data show that tissue neutrophils are perfectly capable of forming NETs in vitro, both in response to MSU crystals and PMA, but that transmigration from blood to tissues does not prime neutrophils regarding MSU- or PMA-triggered NET formation.

## 3. Discussion

Neutrophils mainly exert their actions in tissues beyond circulation and the interactions between neutrophils and MSU crystals are more likely to take place in the tissues, and not in the bloodstream. When neutrophils transmigrate from blood to tissue, they undergo a number of different changes related to this process, e.g., changes in their expression of surface receptors [5]. These changes in the flora of surface receptors will make tissue neutrophils different from blood neutrophils, they become hyper-responsive and prepared to combat microbes. This pre-activated state is also referred to as a primed state. Our results show that (resting) blood neutrophils stimulated with MSU crystals produce ROS and that this oxidative response is highly increased in tissue neutrophils derived from skin chambers. Tissue neutrophils derived from synovial fluid display varying levels of priming, but are clearly not as primed as those from skin chambers. This is in line with our earlier findings that recruitment of neutrophils to joint fluid does not by default induce a primed state [33]. However, the synovial fluid samples used were obtained from patients with non-gouty arthritis and it is possible that synovial fluid neutrophils from gout patients would behave differently. Our findings that in vivo transmigrated neutrophils displayed an increased ROS production were possible to mimic in vitro by incubation with the frequently used priming agent TNFα and also with the cytokine IL1β that play an important role in orchestrate gouty inflammation. In vitro priming is a rapid process (20 min) and the hyper-responsiveness is thus likely not dependent on de novo protein expression, but rather on the exposure of granule-stored receptors. The identity of the surface receptors responsible for the MSU crystal induced responses are still unknown, but based on our data we would predict that they are stored in granules and become accessible after degranulation.

The ROS production of neutrophils triggered by MSU crystals is not extracellular, but strictly took place at intracellular sites. Intracellular ROS production in neutrophils can occur in the phagosome, but also in granules, independent of phagocytosis [7,8]. When pretreating cells with a blocker of actin polymerization, the MSU triggered ROS production was significantly decreased which implies that the oxidative response is dependent on neutrophils (at least trying to) phagocytose the crystals. Sil et al. [4] showed that neutrophils try to, rather than actually manage to, phagocytose the crystals, since the crystals often are bigger than the cells. However, in case such frustrated phagocytosis underlies ROS production, the expected result would be substantial amounts of extracellular ROS escaping from the non-fused phagosome. We did not find any evidence of ROS released extracellularly in response to MSU. Interestingly, colchicine (a drug that inhibits microtubule polymerization and that is used to treat gout) also significantly decreases MSU crystal induced ROS production, but without dampening the response triggered by phagocytosis of opsonized *S. aureus*. This may indicate that the MSU-triggered ROS response is different from that triggered by phagocytosis. The overall knowledge of when, where or how intracellular ROS production occurs is yet limited and will most likely increase as better tools for the study of ROS production become available. 

Neutrophils can be triggered to form NETs by different stimuli including MSU crystals [12,13] and NETs have been found in acutely inflamed joints of gout patients [4,22,25,26,27], as well as in tophi, the uninflamed, chalky deposits of MSU crystals seen around the joints in chronic gout [22,25]. Shauer et al. [25] showed that, during high neutrophil densities, ROS dependent aggregation of MSU triggered NETs could facilitate resolution of inflammation through the degradation of cytokines and chemokines by NET-associated serine proteases. These findings might explain why acute gouty attacks are self-limiting within 1–2 weeks [35] and also why gouty tophi can be uninflamed despite the existence of MSU crystals. Mechanistically, NET formation is often thought to involve ROS production [18]; the most widely used NET trigger in experimental studies is PMA, and NET formation in response to PMA is critically dependent on ROS production and on the ability of MPO to process these ROS intracellularly [20]. There are different views of the role of ROS in MSU induced NET production [13,22,26]. In our experiments, MSU triggered NET formation was more rapid, and reached higher levels than PMA and was not affected by the presence of the NADPH-oxidase inhibitors. Furthermore, neutrophils from one CGD patient as well as from one individual with complete MPO-deficiency formed NETs in response to MSU crystals. These data clearly indicate that the NET formation in response to MSU is independent of ROS. In additional support of this conclusion, we found that tissue neutrophils formed MSU-triggered NETs similarly to blood neutrophils, despite the fact that the former were highly primed for ROS production. Not much is known about tissue neutrophils and NET formation, but our data clearly show that transmigrated cells are fully capable of NET formation in response to MSU as well as to PMA. However, the kinetics and amplitude of NET formation were rather similar between neutrophils from blood and tissue. This demonstrates that the primed phenotype of tissue neutrophils is not evident for NET formation, at least not when induced by PMA or MSU crystals. 

The data presented demonstrate that MSU crystals induce strictly intracellular ROS production as well as NET formation in human neutrophils from blood and tissues, but that these two responses are not mechanistically linked. The ROS response of tissue neutrophils was markedly primed and, interestingly, colchicine seemed to dampen specifically the ROS response from MSU stimulated cells. Colchicine is a widely used treatment for gout, but its mechanism(s) of action is poorly defined. Since ROS is known to mediate important intracellular signaling events and thereby have the potential to modulate inflammatory reactions [7,36,37], it is possible that the effect that colchicine exerts on neutrophil ROS production contribute to its therapeutic effect on gout.

## 4. Materials and Methods

### 4.1. Reagents

TNFα, luminol, isoluminol, horseradish peroxidase (HRP), phorbol 12-myristate 13-acetate (PMA), diphenyleneiodonium (DPI), Cytochalasin B, and colchicine were all from Sigma-Aldrich. IL1β was from PeproTech, superoxide dismutase (SOD) and catalase were from Worthington. MSU crystals were from InvivoGen, GSK2795039 (GSK) was from MedChem Express, Amplex Red reagent was from Thermo Fischer and Sytox green and ProLong Gold antifade reagent were from Molecular Probes. Krebs Ringer phosphate buffer (KRG; pH 7.3) was used with the addition of Ca^2+^ (1 mM) if indicated. 

### 4.2. Study Subjects

Buffy-coats from healthy blood donors were obtained from the blood bank at Sahlgrenska University Hospital, Gothenburg, Sweden. Healthy volunteers (for skin chamber experiments), one CGD patient, one MPO-deficient individual (previously described in [20]) and patients with non-gouty inflammatory arthritis (identified at the Rheumatology Unit at Sahlgrenska University hospital, Gothenburg) participated after giving informed consent. The patients that we collected synovial fluids from were all non-smoking females, aged between 32 and 74 years, one was untreated and the other two were treated with TNF inhibitors. The study was approved by the regional ethical board of Gothenburg, Sweden.

### 4.3. Cell Preparation

Circulating blood neutrophils, from buffy coats or from peripheral blood samples were isolated essentially as described by Boyum et al. [38]. Briefly, after dextran sedimentation at 1× *g* the suspension was centrifuged on a Ficoll–Paque density gradient, and the remaining erythrocytes were lysed by hypotonic treatment. The neutrophils were washed in KRG and finally resuspended in KRG supplemented with Ca^2+^ and kept on melting ice until use. Synovial fluids were aspirated from swollen knee joints by standard procedures [39] and filtrated through 40 µm nylon Cell Strainers (Falcon, Fischer Scientific, Gothenburg, Sweden). After centrifugation, pellets were washed once in KRG and resuspended in KRG supplemented with Ca^2+^ and stored on melting ice until use. Cells were analyzed by flow cytometry and samples with a neutrophil content <50% (based on forward- and side scatter profiles) were excluded from the study. Skin chamber neutrophils were obtained from healthy volunteers as described [40]. Briefly, skin blisters were created by the application of negative pressure and after removal of the blister roofs, collection chambers filled with autologous serum were adjusted over the non-bleeding lesions. Leukocytes will then leave the circulation and transmigrate into the skin chamber fluid. When collecting the cells from the skin chambers after 20 h, yields were between 5–30 million cells with a purity of >85% neutrophils [41]. 

For all samples of in vivo transmigrated neutrophils (from synovial fluids as well as skin chambers), blood was collected from the same donor at the same time and circulating neutrophils were isolated and analyzed in parallel. For direct comparisons (ROS and NETs assays), cell suspensions were analyzed using flow cytometry and diluted so that equal numbers of neutrophils were compared. For some experiments, isolated blood neutrophils were primed with TNFα (10 ng/mL) or IL1β (100 ng/mL) at 37 °C for 20 min prior to use.

### 4.4. ROS Measurements

#### 4.4.1. Chemiluminescense (CL)

ROS production, at extra- and intracellular sites, was detected with isoluminol- or luminol-amplified chemiluminescense (CL), respectively, as described in detail in [42]. For intracellular measurements, neutrophils (5 × 10^5^ cells/well) in KRG (supplemented with Ca^2+^) were added to a 96-well plate together with the cell-permeable luminol (56 µM) while scavenging all extracellular ROS with SOD (50 U/mL) and catalase (2000 U/mL), enzymes that cannot penetrate the cell membrane. After 5 min equilibration at 37 °C, cells were stimulated with MSU crystals (300 µg/mL, unless otherwise stated), PMA (50 nM), or serum-opsonized bacteria (*Staphylococcus aureus* opsonized with 10% human serum for 20 min at 37 °C) added at a multiplicity of infection (MOI) of 10, in the presence or absence of the following inhibitors or drugs; DPI (10 µM), GSK (20 µM) or cytochalasin B (5 µg/mL) at 37 °C for 5 min prior to use, or colchicine (1 µg/mL) at 37 °C for 20 min prior to use. For extracellular superoxide detection, neutrophils (5 × 10^5^ cells/well) in KRG supplemented with Ca^2+^ were added to a 96-well plate together with membrane-impermeable isoluminol (56 µM) in combination with HRP (4 U/mL) and stimulated with MSU crystals (at indicated concentrations) or PMA (50 nM). The CL activity was recorded in a CLARIOstar plate reader (BMG Labtech, Ortenberg, Germany).

#### 4.4.2. Extracellular H_2_O_2_ Production as Measured by Amplex Red

Neutrophils (1.5 × 10^4^ cells/sample) were incubated in Eppendorf tubes together with Amplex Red reagent (50 µM) in the presence of HRP (1 U/mL) at 37 °C and stimulated with MSU crystals (300 µg/mL) or PMA (50 nM). A discontinuous assay system was used; after 20 min incubation, samples were briefly centrifuged to remove MSU crystals and cells. Supernatants were immediately pipetted into a black 96-well plate and analyzed (excitation 570 nm, emission 585 nm) in a CLARIOstar plate reader.

### 4.5. Quantification of NET Formation—Sytox Green Assay

Sytox Green is a cell membrane impermeable nucleic acid stain that labels DNA and that is frequently used to quantify NET formation [34]. Neutrophils (5 × 10^4^ cells/well) in RPMI (without phenol red) were added to black 96-well plates with Sytox Green DNA stain (1.25 μM) and stimulated with MSU crystals at 300 µg/mL (unless otherwise stated) or PMA (50 nM) in the presence or absence of DPI (10 µM) or GSK (20 µM). Fluorescence was measured in arbitrary fluorescence units (excitation 485, emission 535 nm) in a CLARIOstar plate reader after incubation at 37 °C with 5% CO_2_ at indicated time points.

### 4.6. Visualization of NET Formation

Neutrophils (2.5 × 10^5^ cells) were suspended in RPMI and added to acid washed, poly-lysine-coated coverslips. After stimulation with MSU crystals (300 µg/mL) or PMA (50 nM), the cells were incubated at 37 °C in the presence of 5% CO_2_ for indicated time. Cells were fixed with 4% paraformaldehyde for 20 min at room temperature, permeabilized with acetone and methanol (1:1) for 5 min and then blocked for 30 min using PBS with 10% donkey serum and 2% bovine serum albumin. Immunostaining was performed with an anti-MPO antibody (DAKO; A0398), followed by secondary antibody staining (AF488 conjugated donkey-anti-rabbit). An isotype control antibody was used to ensure specific binding of the MPO antibody. Finally, the coverslips were mounted in ProLong Gold antifade reagent containing DAPI and the cells were imaged with fluorescence microscopy (Olympus BX41).

### 4.7. Statistical Analysis

If not mentioned specifically, statistical significance was calculated by the use of the Wilcoxon matched-pairs signed rank test. The analysis was performed in the GraphPad Prism software (version 8.3.1). Statistical significant differences are expressed in the figures by * ≤0.05 and ** ≤0.01. Ns = not significant.

## Figures and Tables

**Figure 1 ijms-21-03750-f001:**
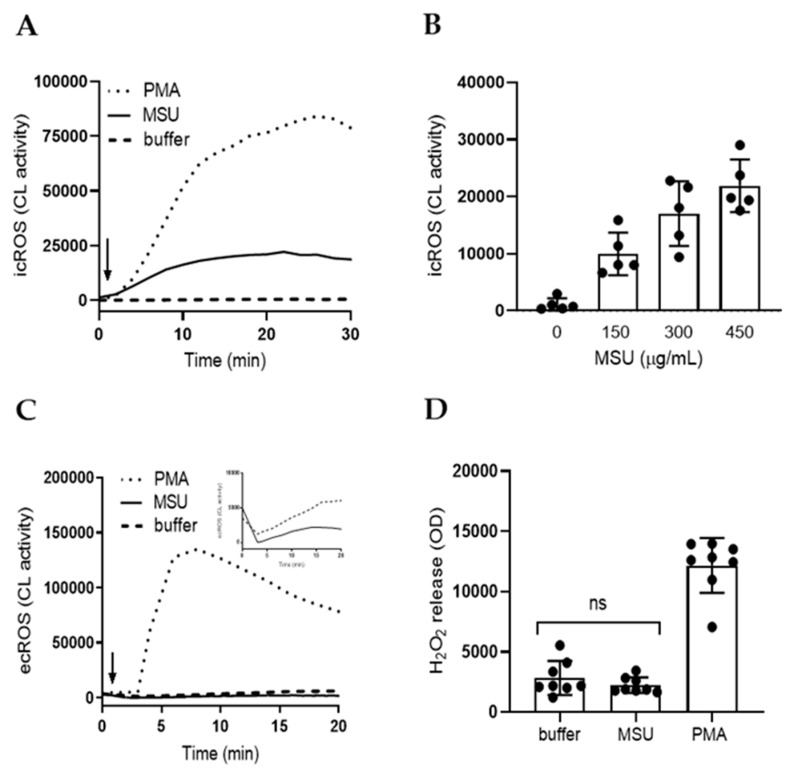
Monosodium urate (MSU) crystals trigger intracellular reactive oxygen species (ROS) production in neutrophils. MSU crystals (300 μg/mL, solid line) triggered significant (*p* < 0.0001 compared to buffer treated cells, *n* = 16), intracellular ROS (icROS) production in neutrophils, as measured by luminol-amplified chemiluminiscense (CL) (**A**), representative kinetic curves are shown), and a clear dose-dependency could be noted when different concentrations of MSU crystals were used. Shown in (**B**) are mean peak CL values +/− SD of five independent experiments. (**C**) A representative kinetic extracellular ROS (ecROS) response, as measured by isoluminol-enhanced CL, of neutrophils stimulated with MSU crystals (500 µg/mL, solid line), PMA (50 nM, dotted line) or buffer (broken line). A close-up of the MSU crystal and buffer traces are also shown in the inset. (**D**) MSU crystals (300 µg/mL) did not trigger extracellular H_2_O_2_ release above buffer-levels, as measured after 20 min incubation with the H_2_O_2_ specific probe Amplex Red. Shown are a mean +/− SD of seven independent experiments. Statistical significance was calculated by the use of the Wilcoxon matched-pairs signed rank test.

**Figure 2 ijms-21-03750-f002:**
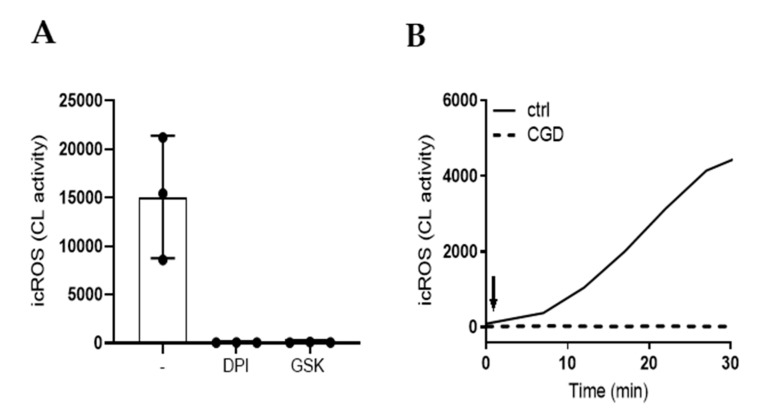
MSU-induced ROS originate from the NADPH-oxidase. (**A**) Neutrophils pre-treated with the NADPH-oxidase inhibitors DPI (10 µg/mL) or GSK (20 µg/mL) did not produce icROS in response to MSU crystals (300 μg/mL); peak values +/− SD are shown (*n* = 3). (**B**) Neutrophils from a patient with CGD (broken line) did not produce icROS in response to MSU crystals (300 µg/mL) as compared to control neutrophils (solid line). Arrows indicate addition of stimuli.

**Figure 3 ijms-21-03750-f003:**
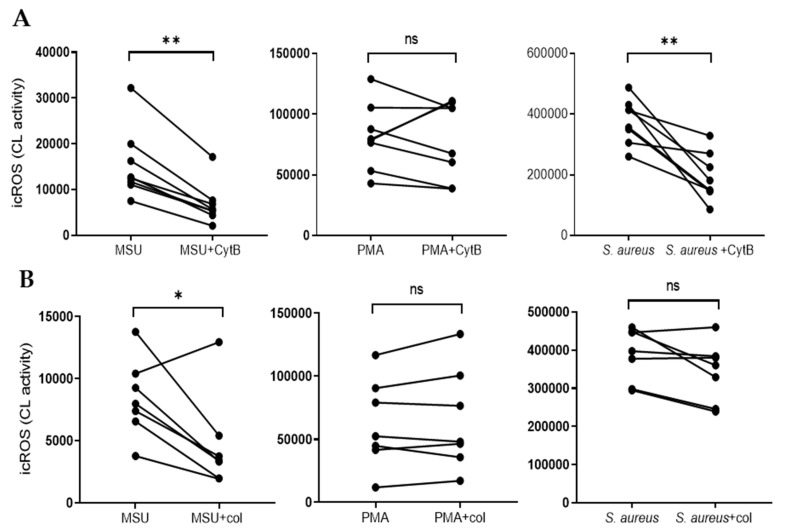
MSU triggered intracellular ROS is inhibited by cytochalasin B (CytB) and colchicine (col). (**A**) Neutrophils were pretreated with cytochalsin B (5 µg/mL), an inhibitor of actin polymerization, and stimulated with either MSU crystals (300 µg/mL), PMA (50 nM) or opsonized *S. aureus* and intracellular ROS production (icROS) was evaluated using luminol amplified CL. Cytochalasin B-treated cells produced lower amounts of icROS in response to MSU crystals and *S. aureus* as compared to untreated cells, while icROS production in response to PMA was not affected by the presence of cytochalasin B. Peak CL values are shown. (**B**) Neutrophils were pretreated with colchicine (1 µg/mL) and stimulated with either MSU crystals (300 µg/mL), PMA (50 nM) or opsonized *S. aureus* and icROS production was evaluated using luminol amplified CL. Colchicine decreased icROS in response to MSU crystals, while icROS production in response to PMA or *S. aureus* was not affected. Peak CL values are shown (*n* = 7). * *p* ≤ 0.05 and ** *p* ≤ 0.01. ns = not significant.

**Figure 4 ijms-21-03750-f004:**
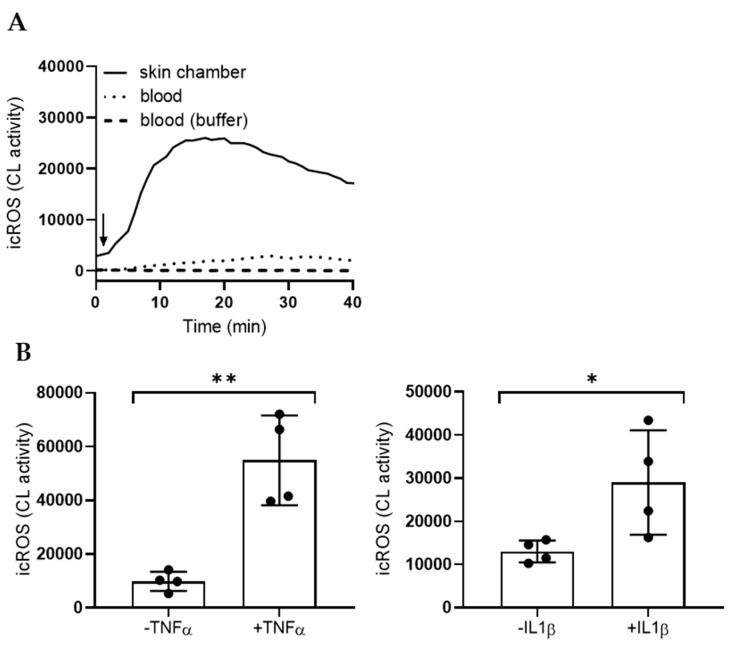
Increased responses to MSU crystals after in vivo transmigration or in vitro priming. (**A**) Tissue neutrophils derived from skin chambers (solid line) produced high amounts of intracellular ROS (icROS) in response to MSU crystals (300 µg/mL) as compared to blood neutrophils (dotted line) from the same donor. A representative kinetic curve out of three independent experiments is shown. Arrow indicates addition of stimuli. (**B**) In vitro priming of blood neutrophils (20 min, 37 °C) with IL-1β (100 ng/mL) or TNFα (10 ng/mL) mimicked in vivo priming; increased icROS responses were seen in the primed neutrophils as compared to resting control neutrophils when treated with MSU crystals (300 μg/mL). Peak values +/− SD are shown and a paired *t*-test was used for statistical comparisons. * *p* ≤ 0.05 and ** *p* ≤ 0.01. ns = not significant.

**Figure 5 ijms-21-03750-f005:**
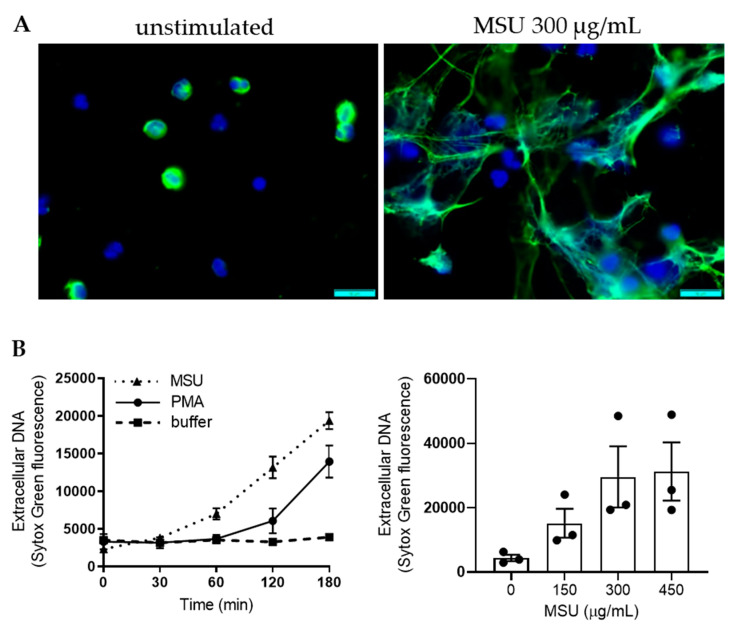
MSU crystals induce neutrophil extracellular trap (NET) formation. (**A**) Neutrophils stimulated with MSU crystals formed NETs as confirmed by DNA- (blue) and MPO- (green) labelled micrographs of neutrophils 3 h after stimulation with MSU crystals (300 µg/mL). Scale bar = 10 µm. (**B**) NET formation as measured with Sytox Green fluorescence over 3 h. To the left, a representative kinetic curve of neutrophils stimulated with buffer (black squares, broken line), MSU crystals (300 µg/mL, triangles, dotted line), or PMA (50 nM, black dots, solid line). MSU crystals (300 µg/mL) triggered a statistically significant DNA release (*p* = 0.0005 compared to buffer treated cells after 180 min, *n* = 12). To the right, NET formation evaluated after 3 h with different concentrations of MSU crystals, mean fluorescence +/− SEM are shown (*n* = 3). Statistical significance was calculated by the use of the Wilcoxon matched-pairs signed rank test.

**Figure 6 ijms-21-03750-f006:**
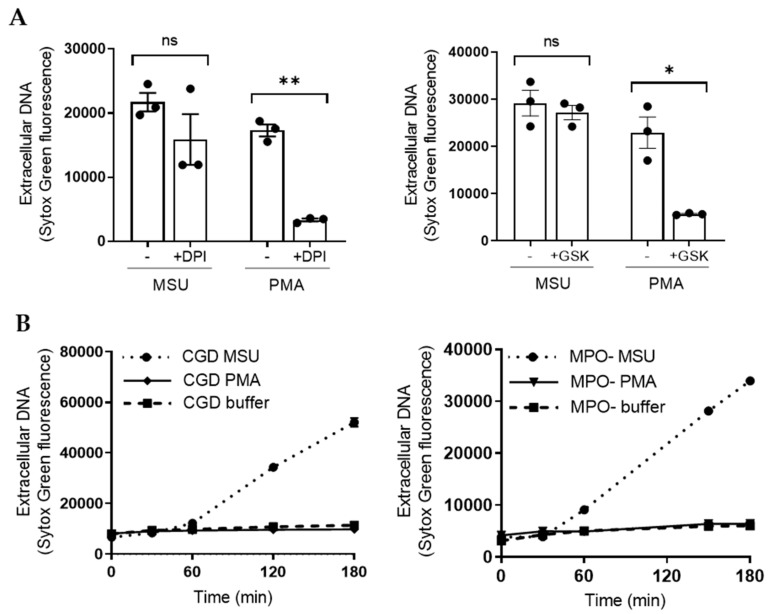
MSU crystal induced NET formation is independent of ROS. (**A**) Neutrophils were pre-treated with the NADPH-oxidase inhibitors DPI (10 μM) or GSK (20 µM) and thereafter stimulated with MSU crystals (300 μg/mL) or PMA (50 nM) before NET formation was evaluated by Sytox green measurement after 3 h. As opposed to PMA stimulated neutrophils, MSU treated neutrophils formed NETs also in the presence of the inhibitors. Mean +/− SEM of three independent experiments are shown, a paired *t*-test was used for statistical comparisons. (**B**) Neutrophils from one patient with CGD (left) as well as from one individual with complete MPO-deficiency (MPO-, right) formed NETs in response to MSU (300 μg/mL), dotted lines, but not to PMA (50 nM, solid lines). * *p* ≤ 0.05 and ** *p* ≤ 0.01. ns = not significant.

**Figure 7 ijms-21-03750-f007:**
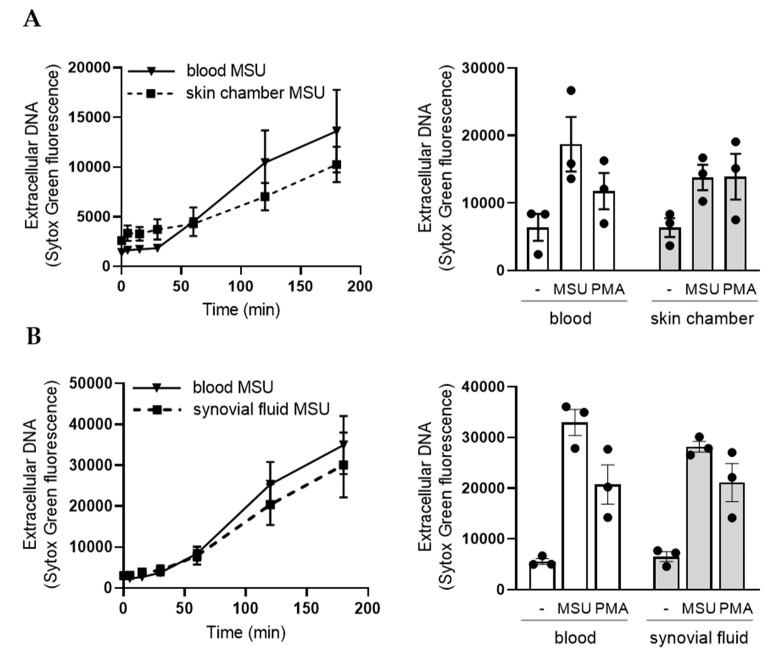
In vivo transmigrated neutrophils are not primed for NET formation. Sytox green fluorescence assay was used to evaluate NET formation of tissue neutrophils from skin chamber fluid (**A**) or synovial fluid (**B**) after stimulation with MSU crystals (300 µg/mL). Tissue neutrophils (broken lines) formed NETs in a similar manner as blood neutrophils (solid lines) from the same donor assayed in parallel. Shown are representative curves for skin chamber neutrophils (**A**), and synovial fluid neutrophils (**B**), and the mean fluorescence values +/− SEM at 3 h from three independent experiments of each type of tissue neutrophils.

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
