# Peer review of "In Vivo Transmigrated Human Neutrophils Are Highly Primed for Intracellular Radical Production Induced by Monosodium Urate Crystals"

_ijms, 2020, doi:10.3390/ijms21113750_

Round 1

Reviewer 1 Report

Include in the figures and figure captions, the symbols of statistical differences, not all have them, although they are mentioned in the text

Author Response

As the reviewer points out, the way our statistical calculations were performed is perhaps not 100% clear. The dose-response figures (Fig. 1B & 5B right panel) are based on limited experimental repeats, just to illustrate the fact that the responses we describe depend on the MSU dose. We did not perform any statistical analyses on the data in those figures. In contrast, the fact that we do see a (statistically significant) response to a fixed MSU dose (300µg/mL) was analyzed statistically and described in the text both regarding ROS production and NET formation. These observations were repeated many more times than the dose-response experiments and to clarify this we have added the information to the figure legends describing that a 300 µg/mL dose of MSU evoke a ROS response (Fig. 1A) and NET formation (Fig 5B left panel). Since these particular figures show representative curves (and technical repeats in Fig. 5B left panel), we have not added any symbols to denote statistics to the figures.

Reviewer 2 Report

In this study, it was demonstrated that transmigrated human neutrophils are primed for (intracellular) ROS and NET formation. This is a well-designed study, interesting topic, and very well written. 

I will have a few minor comments/suggestions. 

1) introduction: major concern is the use of out-dated literature. I would suggest adding or replacing some references by recent literature (<5-10 years). Especially reference 5 is from 1995, reference 7 is from 2007. References citing NETs should also be more up to date since this is a very popular topic. 

2) results

2a) format y-axis of Figure 1A: reduce the major tick interval

2b) abbreviation of CL is not mentioned in the text

2c) Liter is a SI unit and should be capitalized (mL instead of ml)

2d) I would suggest to make a separate figure out of Figure 1E and 1F. 

2e) add a subheading to the paragraph starting at page 4 (discussing figure 2) 

2f) NET quantification data are presented as ''extracellular DNA'' however, it should be stated which units are presented (in m&M or figure legends). assuming this is MFI. 

2g) all results are very well described and easy to follow by reading the text. however, figure 6 is described by only two sentences (page 7). I would suggest elaborating more on these findings, explicitly also describing fig.6a and 6b separately, since these are of high importance. 

3) materials and methods - no comments

4) discussion - in general, very well written but again citing out-dated literature. please add more recent literature. 

Figure 6 and Supplementary Figure 1 shows variability in different donors. could you provide (clinical) characteristics of your donors? age, gender, ethnicity, smoking, medication, BMI are examples of characteristics which could be of importance for your interpretation of your variable results. for example, it is known that smoking negatively affects PMN chemotaxis, ROS production, and NET formation. In case these characteristics are different for these subjects, it would be needed to discuss this in the discussion part.

Author Response

Thank you for reviewing our manuscript and for your remarks.

1 & 4) We agree and have updated the 2 old references (5 and 7) and also added a new reference (ref nr 12; Boeltz et al 2019) 

2 a-e) We agree and have made the following changes to the revised manuscript:

  • Y-axis of Fig. 1A has been reformatted
  • The abbreviation CL is mentioned more clearly in the text
  • ml has been changed to mL throughout
  • 1 has been split into two figures and a new sub-heading has been introduced to the results section

2f) The units presented are arbitrary fluorescence units and we have added this information to the M&M section.

2g) We have changed the text in the results section so that figures 6a & 6b (now 7a & 7b) are described separately.

Last point) Regarding our experiments on tissue neutrophils; neutrophils derived from skin chambers were from healthy donors. Synovial fluid samples were collected from 3 individual patients that all suffered from non-gouty arthritis. All patients are female non-smokers.

It is correct that various patient characteristics could potentially influence neutrophil behavior in vitro, but from our very limited group of patients we could not find any obvious differences that could account for the different priming status (Supl. Fig 1). We have added additional patient characteristics to the revised manuscript.